# Proliferation to Apoptosis Tumor Cell Ratio as a Biomarker to Improve Clinical Management of Pre-Malignant and Symptomatic Plasma Cell Neoplasms

**DOI:** 10.3390/ijms22083895

**Published:** 2021-04-09

**Authors:** María A. Vasco-Mogorrón, José A. Campillo, Adela Periago, Valentin Cabañas, Mercedes Berenguer, María C. García-Garay, Lourdes Gimeno, María F. Soto-Ramírez, María D. Martínez-Hernández, Manuel Muro, Alfredo Minguela

**Affiliations:** 1Immunology Service, Clinic University Hospital Virgen de la Arrrixaca (HCUVA), Biomedical Research Institute of Murcia (IMIB), 30120 Murcia, Spain; atlantisady@gmail.com (M.A.V.-M.); josea.campillo@carm.es (J.A.C.); lourdes.gimeno@carm.es (L.G.); mariaf.soto@carm.es (M.F.S.-R.); lola.mtz.hdz@gmail.com (M.D.M.-H.); manuel.muro@carm.es (M.M.); 2Hematology Service, General University Hospital Rafael Méndez, Biomedical Research Institute of Murcia (IMIB), 30813 Murcia, Spain; adelam.periago@carm.es; 3Hematology Service, Clinic University Hospital Virgen de la Arrrixaca (HCUVA), Biomedical Research Institute of Murcia (IMIB), 30120 Murcia, Spain; valentin.cabanas@carm.es (V.C.); mcarmen.garcia35@carm.es (M.C.G.-G.); 4Hematology Service, General University Hospital Santa Lucía, Biomedical Research Institute of Murcia (IMIB), 30202 Murcia, Spain; mercedes.berenguer@carm.es; 5Human Anatomy Department, Medicine Faculty, Biomedical Research Institute of Murcia (IMIB), Murcia University, 30120 Murcia, Spain

**Keywords:** plasma cell neoplasm, MGUS, SMM, MM, treatment, proliferation, apoptosis

## Abstract

Proliferation and apoptosis of neoplastic cells are prognostic biomarkers in plasma cell neoplasms (PCNs). The prognostic capacity of proliferation to apoptosis ratio (Ratio-PA) in the era of immunomodulatory treatments is re-evaluated in 316 gammopathy of undetermined significance (MGUS), 57 smoldering multiple myeloma (SMM), and 266 multiple myeloma (MM) patients. Ratio-PA of 0.77 ± 0.12, 1.94 ± 0.52, and 11.2 ± 0.7 (*p* < 0.0001) were observed in MGUS, SMM, and MM patients. Ten-year overall survival (10y-OS) rates for patients with low/high Ratio-PA were 93.5%/77.3% *p* < 0.0001) for MGUS, 82.5%/64.7% (*p* < 0.05) for SMM, and 62.3%/47.0% (*p* < 0.05) for MM. For patients with low, intermediate, and high risk, 10y-OS for low/high Ratio-PA were 95.5%/72.9% (*p* < 0.0001), 74.2%/50.4% (*p* < 0.0001), and 35.3%/20.0% (*p* = 0.836), respectively. Ratio-PA was an independent prognostic factor for OS (HR = 2.119, *p* < 0.0001, Harrell-C-statistic = 0.7440 ± 0.0194) when co-analyzed with sex, age, and standard risk. In patients with Ratio-PA^high^, only first-line therapy with VRd/VTd, but not PAD/VCD, coupled with ASCT was associated with high 10y-OS (82.7%). Tumor cell Ratio-PA estimated at diagnosis offers a prognostic biomarker that complements standard risk stratification and helps to guide the clinical management of pre-malignant and symptomatic PCNs. Every effort should be made to provide first-line therapies including VTd or VRd associated with ASCT to patients with Ratio-PA^high^ at higher risk of progression and death.

## 1. Introduction

Multiple myeloma (MM) is a plasma cell neoplasm (PCN) with a relapsing clinical course requiring several lines of therapy in most patients. Nonetheless, novel standard first-line treatments combining immunomodulatory drugs (lenalidomide, thalidomide, or pomalidomide, -IMiDs-), proteasome inhibitors (bortezomib, Carfilzomib, or Ixazomib, -PIs-), and monoclonal antibody as daratumumab and tandem autologous stem cell transplantation (ASCT) have increased the rate of complete response (CR) and prolonged treatment-free and survival periods [1]. Unfortunately, not all patients are eligible for ASCT, so they have to receive treatments that, although improved in recent years, are not as effective as those associated with ASCT. Nonetheless, regardless of the type of treatment, the disease displays substantial clinical heterogeneity in presentation and course, thus underlining the need for biomarkers that allow us to adapt the therapy not only to the patient’s biological and clinical conditions, but also to the real risk of the disease. Currently, there are several risk stratification systems depending on the type of PCN, from premalignant monoclonal gammopathy of undetermined significance (MGUS) [2,3,4] and smoldering MM (SMM) [2,3,5,6,7] to the symptomatic MM [8]. MGUS is the most common PCN, with a prevalence of 3.2% in the general population older than 50, and increasing with age [9,10,11]. Although the progression rates to MM in asymptomatic MGUS and SMM are about 1% and 10% per year, respectively [11,12], prognostic biomarkers are also needed in MGUS and SMM for counseling, clinical care, and follow-up, and for the design of clinical studies in patients at high risk [9].

In MM, biochemical and genetic prognostic factors such as albumin, beta-2 microglobulin (b2m), lactate dehydrogenase (LDH), and the high-risk cytogenetic abnormalities del(17p) and t(4;14) detected by fluorescent in situ hybridization (FISH) have been incorporated in the Revised International Staging System (RISS), to provide a powerful prognostic tool to risk-stratify patients at diagnosis [8]. However, the presence of high-risk cytogenetics and the concurrence of elevated levels of LDH and b2m are rarely observed at the onset of the disease; therefore, RISS generally underestimates high-risk and overestimates intermediate-risk patients and, as a consequence, patients at intermediate risk show notable clinical variability, indicating that some of these patients could have benefited from more effective first-line treatments [13].

The myelomatous plasma cell (PC) proliferation index provides insight into the biology of the neoplastic cell, and it is a strong prognostic marker in active MM [14,15,16] and SMM [17,18] and even in MGUS [19]. However, the expansion of myeloma neoplastic clone is determined by a balance between proliferation on the one hand, and induction or blockade of apoptosis on the other hand. In fact, myeloma is characterized by very slow proliferation of malignant PC, suggesting that pathogenesis and resistance to treatment of these cells within the bone marrow (BM) may be related to resistance to apoptosis [20]. Imbalances in the expression of the Bcl-2 family members (higher expression of the antiapoptotic Bcl-2 and Mcl-1, and lower of the proapoptotic Bax [21]) result in defects in programmed cell death, which are associated with malignancy, aggressiveness, and chemoresistance of tumors [20]. Various therapeutic modalities that are effective in MM modulate levels of the Bcl-2 family members, the expression of which is primarily regulated by p53, nuclear factor κB, and STAT factors [20]. Both IMiDs and PIs inhibit the NF-κB cell survival pathway and stimulate apoptosis of malignant PCs [22,23]. Besides, PIs stimulate the expression of the proapoptotic Fas ligand/receptor members [23].

In contrast with solid tumors, where higher proliferation rates are associated with increased apoptosis of neoplastic cells, [24] in PCNs, an inverse relationship between PC proliferation and apoptosis rates has been well-established [25]. Nonetheless, an elevated proliferation to apoptosis ratio (Ratio-PA) of BM-PC was an independent prognostic factor associated with shorter patient’s overall survival (OS) both in solid tumors (cervical adenocarcinoma [26] and glioblastoma multiforme [27]) and in MM [14,15,16]. In MGUS and SMM, high Ratio-PA of BM-PC has been associated with active disease and progression, [28,29] but its prognostic value in patient’s OS has been poorly investigated.

This study shows how the combined analysis of BM-PC proliferation and apoptosis rates, easily estimated by flow cytometry, complements standard risk stratification and increases their prognostic capacity to identify high-risk MGUS patients, most of whom will progress or die in the first few years after diagnosis. Besides, it also contributes to identification of SMM and risk-I/-II MM patients who, due to their higher risk of progression and death, could benefit from the most effective therapies reserved for MM patients with high-risk cytogenetics. The prognostic capacity remained valid even in MM patients who relapsed after stringent CR (sCR) with negative minimal residual disease (MRD), and therefore, it would be equally useful for the most effective drugs used in current clinical practice.

## 2. Results

### 2.1. Patient Characteristics

Table 1 presents biological, clinical, and therapeutic characteristics of patients. Ten-year progression-free survival (10y-PFS) and OS (10y-OS) rates were 86.4% and 90.8% for MGUS, 50.0% and 75.0% for SMM, and 40.6% and 54.5% for MM. According to the standard risk stratification (risk-I, -II, and -III), MGUS showed 10y-PFS of 93.8%, 78.4%, and 50.0% (*p* < 0.0001) and 10y-OS of 95.3%, 85.6%, and 62.5% (*p* < 0.0001); SMM showed 10y-PFS of 58.3%, 44.4%, and 20.0% (*p* < 0.05) and 10y-OS of 87.5%, 70.4%, and 20.0% (*p* < 0.01); and MM showed 10y-PFS of 44.4%, 36.1%, and 13.3% (*p* < 0.0001) and 10y-OS of 79.6%, 47.4%, and 16.7% (*p* < 0.0001). According to the type of treatment, 10y-PFS and 10y-OS rates were 26.8% and 40.2% for No-ASCT (*p* < 0.05), 35.5% and 66.1% for ASCT with PAD or VCD, and 60.7% and 85.5% for ASCT with VTd or VRd (Figure 1).

### 2.2. Prognostic Value of BM-PC Proliferation to Apoptosis Ratio (Ratio-PA)

First we analyzed the biological characteristics of PCs infiltrating the BM in MGUS, SMM, and MM. Increasing total BM-PCs in the histology analysis (4.72 ± 0.16%, 15.9 ± 1.5% and 3.5 ± 1.6%, *p* < 0.0001) and BM-PC proliferation rates (1.86 ± 0.1%, 2.17 ± 0.23% and 3.26 ± 0.19%, *p* < 0.0001), but decreasing apoptotic rates (9.53 ± 0.5%, 6.41 ± 1.1, and 4.34 ± 0.39, *p* < 0.0001) were observed for MGUS, SMM, and MM patients (Figure 2A). A proliferation-to-apoptosis ratio (Ratio-PA) of BM-PC was calculated for each patient and analyzed in MGUS, SMM, and MM stages (0.77 ± 0.12, 1.94 ± 0.52, and 11.2 ± 0.7, *p* < 0.0001; Figure 2B).

The prognostic capacity for PFS of BM-PC proliferation and apoptosis rates and Ratio-PA were evaluated by ROC analysis in total PCN patients (Figure 2C). Since Ratio-PA showed higher area under the curve (AUC = 0.72) than proliferation (AUC = 0.61) and apoptosis (AUC = 0.64) rates, cutoffs for Ratio-PA with the highest prognostic capacity for PFS were estimated for MGUS (cutoff = 0.91, sensitivity = 55.5% and specificity = 82.1%), SMM (cutoff = 1.27, sensitivity = 51.1% and specificity = 87.1%), and MM (cutoff = 1.15, sensitivity = 62.5% and specificity = 65.2%). Finally, the proportion of patients over their correspondent cutoffs (Ratio-PA^high^) were estimated for MGUS, SMM, and MM (16.8%, 29.8% and 50.8%, *p* < 0.0001).

Next, we explored the prognostic capacity of the Ratio-PA for PFS and OS of MGUS, SMM, and MM patients by Kaplan-Meier analysis (Figure 3A). The 10y-PFS and 10y-OS rates for patients with Ratio-PA^high^ vs. Ratio-PA^low^ were 90.1% vs. 67.9%% (*p* < 0.0001) and 93.5% vs. 77.3% (*p* < 0.0001) for MGUS, 57.5% vs. 35.3% (*p* = 0.078) and 82.5% vs. 64.7% (*p* < 0.05) for SMM, and 49.2% vs. 32.1% (*p* < 0.05) and 62.3% vs. 47.0% (*p* < 0.05) for MM, respectively.

Cox regression analysis of total PCN showed that Ratio-PA was an independent prognostic factor for PFS (HR = 2.289, *p* < 0.0001, Harrell C-statistic = 0.7104 ± 0.0193) and OS (HR = 2.119, *p* < 0.0001, Harrell C-statistic = 0.7440 ± 0.0194) when analyzed together with sex (shorter PFS and OS for men than women), age (shorter PFS and OS for elders), and standard risk stratification (HR = 2.247, *p* < 0.0001 for PFS and HR = 3.280, *p* < 0.0001 for OS; Figure 3B). Comparable HR results were observed when MGUS, SMM, and MM stages were analyzed separately (see Figure 3C for details).

### 2.3. Ratio-PA Complements the Prognostic Capacity of Standard Risk Stratification in PCNs

The Ratio-PA showed prognostic capacity in pre-malignant and symptomatic PCNs not only by itself, but also complementing the standard risk stratification estimated in each type of PCN following current standard criteria, especially for low- (Risk-I) or intermediate- (Risk-II) risk patients. According to the low/high Ratio-PA, we observed 10y-PFS and 10y-OS rates of 85.4%/47.9% (*p* < 0.0001) and 95.5%/72.9% (*p* < 0.0001) for low-risk patients; 65.7%/35.4% (*p* < 0.0001) and 74.2%/50.4% (*p* < 0.0001) for intermediate-risk patients; and 23.5%/16.0% (*p* = 0.703) and 35.3%/20.0% (*p* = 0.836) for high-risk patients (Figure 4, left plots). It is noteworthy that 24.0% of standard low-risk patients who had Ratio-PA^high^ showed PFS and OS curves close to those seen in intermediate-risk patients; and in the same way, 35.2% of intermediate-risk patients who had Ratio-PA^high^ showed PFS and OS curves that were closed to those seen in high-risk patients.

Similar results were observed for each type of PCN. According to the low/high Ratio-PA, MGUS, SMM, and MM patients showed 10y-PFS rates of 95.7%/83.3% (*p* < 0.05), 68.4%/33.3% and 58.6%/48.6% for sRisk-I; 84.2%/60.0% (*p* < 0.01), 50.0%/44.4% and 47.1%/29.1% (*p* < 0.05) for sRisk-II, and 60.0%/45.0%, 33.3%/10.0% and 25.0%/10.5% for sRisk-III, respectively; and 10y-OS rates were 97.1%/88.8%, 94.7%/66.6% (*p* < 0.01) and 93.1%/74.2% (*p* < 0.05) for sRisk-I, 89.5%/73.3% (*p* < 0.05), 77.8%/66.7% and 55.2%/43.0% for sRisk-II, and 80.3%/60.1%, 33.3%/20.0% and 25.0%/10.5% for sRisk-III, respectively (Figure 4, right plots).

### 2.4. Delays in BM Sample Processing Reduces the Prognostic Capacity of the Ratio-PA

Given that both proliferation and apoptosis rates of BM-PC are functional biological properties, we evaluated the impact that delays in BM sample processing could have in the prognostic capacity of the Ratio-PA. Ratio-PA (3.7 ± 0.54, 2.45 ± 0.57, and 1.1 ± 0.23) and the proportion of patients with Ratio-PA^high^ (39.7%, 31.3%, and 19.7%) decreased proportionally to the time elapsed from extraction (2 h, 4 h, and 24 h, respectively) (Figure 5A). Differences in the distribution of patients by PCN stages or standard risk groups for samples processed at different times, which could have justified the differences observed in their Ratio-PA, were ruled out (Figure 5B). Nonetheless, patients with Ratio-PA^high^ showed lower PFS rates than patients with Ratio-PA^low^ for samples processed in 2 h (81.3% vs. 39.5%, *p* < 0.0001), 4 h (68.3% vs. 42.8%, *p* < 0.05), or 24 h (70.9% vs. 48.3%, *p* < 0.01), although differences in the PFS and its statistic signification decreased in inverse proportion to the time elapsed until laboratory processing. Besides, Cox regression analysis confirmed that Ratio-PA was an independent prognostic factor in the PFS of PCN patients, but again with decreasing HR (3.189, 1.865, and 1.890), level of statistic signification (*p* < 0.0001, *p* < 0.01, and *p* < 0.05), and the Harrell C-statistic (0.7073 ± 0.0292, 0.6591 ± 0.0328, and 0.6728 ± 0.0402) as time until processing increased (Figure 5C).

### 2.5. Tandem ASCT with VTd or VRd Are the Most Effective First-Line Treatments for Patients with Ratio-PA^high^

Finally, to explore which therapies of those used in current clinical practice were the most effective in patients with Ratio-PA^high^, we evaluated the 10y-OS rate and the relapse rate after sCR in symptomatic MM (*n* = 254), as well as in MGUS (*n* = 19) or SMM (*n* = 24) patients that progressed and required treatment (*n* = 297 total). Patients were classified according to the first-line treatment used (Figure 6A). In patients eligible for ASCT, Ratio-PA^low^ was associated with good 10y-OS rates (77.4% vs. 87.5%) for both PAD/VCD and VRd/VTd treatments, whereas in patients with Ratio-PA^high^, only VRd/VTd, but not PAD/VCD, were associated with high 10y-OS rates (82.7% vs. 55.2%). Patients not eligible for ASCT showed much lower 10y-OS rates, while non-significant differences were observed according to the Ratio-PA^low/high^ (46.1% vs. 35.6%).

Higher 10y-OS rates observed in patients with Ratio-PA^high^ treated with ASCT and VRd/VTd were related to a lower relapse rate than the one for patients treated with ASCT and PAD/VCD or No-ASCT (27.8% vs. 73.7% and 78.3%, *p* < 0.01). In the same line, high 10y-OS rates observed in patients with Ratio-PA^low^ treated with ASCT, either with PAD/VCD or VRd/VTd, were associated with low relapse rates, 37.5% and 44.4% respectively, compared to the relapse rate of patients not eligible for ASCT (63.6%) (Figure 6B).

Patients who did not relapse in the first six months after sCR showed higher 10y-OS rates compared to those who relapsed (86.0% vs. 51.6%, *p* < 0.0001). As expected, the Ratio-PA (low/high) did not have a significant impact on the 10y-OS rates of patients who did not relapse (87.0% vs. 84.6%); however, Ratio-PA^low^ identified patients who, after relapse, showed higher 10y-OS rates than those with Ratio-PA^high^ (72.7% vs. 41.0%, *p* < 0.05) (Figure 6C).

To further explore the impact of Ratio-PA in the outcome of patients requiring anti-myeloma treatment, Appendix A shows the impact of Ratio-PA according to their eligibility for ASCT or their relapse status in the RISS stages. In ASCT-ineligible patients, only those with RISS-I and Ratio-PA^high^ showed lower 10y-OS. In ASCT-eligible patients, RISS-I, RISS-II, and RISS-III stages with Ratio-PA^high^ showed lower 10y-OS. In patients without relapse, Ratio-PA^high^ did not lead to differences in the 10y-OS in patients with RISS-I, but the survival rate was lower in those with RISS-II and RISS-III. In relapsed patients with RISS-I and RISS-II, the Ratio-PA^high^ led to lower 10y-OS. Nonetheless, these results should be confirmed in larger series due to the reduced number of patients in some subgroups.

## 3. Discussion

Plasma cell neoplasms with a prevalence >3% on population aged ≥50 (>5% on aged >70) for MGUS [9,10] and close to 1% for MM are one of the most common pathologies in hematology clinics worldwide. Although most of them are asymptomatic, 1% yearly evolves to a symptomatic PCN that requires treatment, as do patients diagnosed of MM. Fortunately, effective risk stratification methods revised in recent years [7,8,30] have significantly contributed to rationalizing clinical follow-up and therapeutic management of these patients. However, substantial clinical heterogeneity in presentation and course is observed among patients within the same risk status or under comparable therapy regimens, underlining critical differences in neoplastic PC biology and its resistance to current anti-myeloma drugs. This study, performed in a large series of patients under current standard treatments and followed-up for a period of 10 years (mean ± SD, 55.5 ± 39 months), shows that proliferation and apoptosis rates of the neoplastic cells significantly determined the progression of the disease and the survival of patients with asymptomatic PCNs and also with symptomatic MM that required treatment. As suggested by our data, the net growth of the tumor estimated as a Ratio-PA of the BM-PCs could determine the aggressiveness of the disease and its resistance to current treatments. Our results also reveal two issues of crucial relevance for the clinical management of these patients: firstly, patients with high net tumor growth (Ratio-PA^high^) will respond mostly to first-line therapies including VTd or VRd associated with ASCT, while PCNs with low net tumor growth will show comparable good results with first-line therapies including PAD/VCD or VTd/VRd associated with ASCT; and secondly, proliferation and apoptosis of neoplastic PC should be analyzed in the shortest possible time from BM extraction (preferably <2 h), since time and/or transport alter these functional characteristics of the tumor cell, reducing the prognostic capacity of the Ratio-PA and consequently worsening the correct decision-making in both premalignant and symptomatic PCNs.

Indeed, delays in BM sample processing seemed to affect the prognostic capacity of Ratio-PA. Nonetheless, its independent predictive value remained valid, although it decreased, even in samples that were analyzed 24 h after extraction. This is in agreement with numerous studies in PCNs demonstrating a prognostic capacity of either the proliferation rate, [14,15,16,17,18,19] the apoptosis rate [20], or Ratio-PA [28,29]. Although, in light of our results, these studies may have underestimated the prognostic capacity of these parameters, it has been well-established that the PC proliferation rate is one of the most important independent prognostic factors, a specific marker of MM aggressiveness [31,32,33] and an early marker for disease progression and death [31,34]. However, our data suggest that the inverse relationship between BM-PC proliferation and apoptosis and particularly the balance established between these two parameters is what would ultimately determine the expandability and aggressiveness of the neoplastic PC, and would explain why Ratio-PA was the parameter with the highest predictive value in our PCN series. In fact, immunomodulatory therapies base their anti-myeloma action on the induction of PC apoptosis, which could help to explain why ASCT eligible patients treated with IMiDs showed the best outcome in our series [22,23]. It has been reported that the neoplastic population in MM consists of three clonally-related compartments [35]: (1) self-maintaining stem-cells, (2) growth fraction, and (3) differentiated non-proliferative PCs that will die by apoptosis and necrosis. High Ratio-PA would be indicative of an imbalance between these compartments, in favor of the proliferative one, which would justify a greater aggressiveness and a higher progression rate in these patients.

Improvement of clinical outcomes in MM with the introduction of immunomodulatory drugs has caused some biomarkers to lose their prognostic value, and has made it necessary to redefine the stratification criteria to adjust their predictive capacity [7,8,30]. However, the clinical variability observed among patients with the same risk makes it necessary to complement the standard risk stratification with biologically meaningful biomarkers also reassessed for current therapies. In this study, the prognostic capacity of the Ratio-PA was re-evaluated for current treatments in MM, and it was found that not only does it continue to maintain its independent prognostic capacity, but it is also capable of complementing the RISS. Ratio-PA^high^ identified 24.0% of low standard risk patients who showed rates of PFS and OS similar to patients with intermediate risk. Furthermore, Ratio-PA^high^ identified 35.2% of intermediate-risk patients whose survival curves were closed to those at high standard risk. These patients could have benefited from first-line treatments reserved for high-risk patients. Moreover, if eligible for ASCT, these patients should receive regimens based on VTd or VRd associated to ASCT, the only ones that in our series seem to be effective against PC neoplasms with a high net tumor growth.

Unfortunately, Ratio-PA was unable to provide additional information on the prognosis of patients not eligible for ASCT, since all of them presented an unfavorable clinical course irrespective of the proliferation or apoptosis rates of their neoplastic cells. An exception was ASCT non-eligible patients with RISS-I, where Ratio-PA^high^ identified a subset of patients at higher risk. However, a Ratio-PA^low^ at diagnosis would point at 36% of cases that relapsed after sCR and then had prolonged survival. In contrast, Ratio-PA^high^ predicted a cohort of patients with much poorer OS who could benefit from consolidation therapy, tandem ASCT, or intensive maintenance after relapse [36]. Recently, it has been described that evaluating PC proliferative index post-transplant is a powerful predictor of prognosis in myeloma patients failing to achieve a complete response [37]. Nonetheless, our data suggest that this is a characteristic that could be fixed from the diagnosis. Finally, Ratio-PA^high^ did not impact the good clinical course of patients with RISS-I who maintained sCR beyond 6 months post-ASCT, although it could be pointing at RISS-II and RISS-III patients with worse clinical course, even though they maintained early sCR post-ASCT.

Although it should be confirmed in larger series, our results suggest that Ratio-PA could be helpful in the clinical management of SMM patients. Excluding ultra-high-risk SMM patients (risk of progression >80% in the first two years) who should be diagnosed as symptomatic MM [38,39], the optimal time to treat SMM patients remains controversial [40,41,42]. In our series, Ratio-PA^high^ identified 29.8% of SMM patients with 35.3% 10y-PFS who could have benefited from an early treatment, [43] since safer and more effective therapies are now available. Watchful waiting was appropriate in an era of only alkylating agents and corticosteroids, but is no longer justifiable for these high-risk patients [39].

Ratio-PA analysis could also facilitate risk-adapted follow-up and clinical management in MGUS. MGUS progression to malignant PCNs occurs at a rate of 1% per year [11]. However, in our series, Ratio-PA^high^ pointed out patients with a risk of progression and death 3.2 and 3.4 times higher, respectively. Shorter OS of MGUS patients with Ratio-PA^high^ supports the possibility of a direct negative effect of the monotypic clone inducing severe organ damage, such as by favoring thrombosis, infection, or osteoporosis, amongst others; [44] therefore, it would be reasonable to assess the possibility of early treatment in these high-risk MGUS cases.

Altogether, these results show that the proliferation-to-apoptosis ratio (Ratio-PA) of PC, estimated at diagnosis by flow cytometry in the shortest possible time from BM extraction (preferably <2 h), offers a prognostic biomarker that complements standard risk stratification and helps to guide the clinical management of pre-malignant and symptomatic PCNs. Our results also show that every effort should be made to provide first-line therapies including VTd or VRd associated with ASCT to patients with Ratio-PA^high^, at higher risk of progression and death. Finally, Ratio-PA^high^ predicted a cohort of patients who could benefit from consolidation therapy, tandem ASCT, or intensive maintenance when they relapse after stringent complete response.

## 4. Patients and Methods

### 4.1. Patients and Samples

EDTA anti-coagulated BM samples were obtained at diagnosis from 639 consecutive patients with PCNs in regular clinical practice from seven hospitals in the Region of Murcia, Spain. BM samples were also obtained for MRD assessment at three and six months after ASCT, and under the suspicion of loss of sCR. Patients were enrolled between 2010 and 2017, and followed-up until February 2021. This study was approved by the Research Ethics Committee Institutional Review Board (IRB-00005712). Written informed consent was obtained from all patients in accordance with the Declaration of Helsinki.

Following the IMWG criteria [8], patients were classified in 316 MGUS, 57 SSM, and 266 MM. Standard risk stratification (sRisk) in MGUS [30], SMM [7], and MM [8] was done following updated criteria. Patients were grouped as low (Risk-I), intermediate (Risk-II), and high (Risk-III) risk. In MGUS, disease progression was computed when it progressed to SMM or MM. In SMM and MM progression, complete response (CR) and relapse were estimated following the IMWG Uniform Response Criteria for Multiple Myeloma [39,45]. Treatments and management were at the discretion of the hematologists based on patient condition and tumor risk. Briefly, conventional first-line therapy for patients not eligible for ASCT included bortezomib, melphalan, and prednisone (VMP), bortezomib and dexamethasone (Vd), or more recently, lenalidomide and dexamethasone (Rd). In ASCT-eligible patients, first-line therapy included bortezomib, cyclophosphamide, dexamethasone (VCD) or bortezomib, doxorubicin, and dexamethasone (PAD) or more recently bortezomib, thalidomide, and dexamethasone (VTd) or bortezomib, lenalidomide, and dexamethasone (VRd), and ASCT conditioning with melphalan 200 mg/m^2^ (dose ranging from 200 to 100–140 mg/m^2^ in the case of renal impairment).

### 4.2. Immunophenotyping, MRD, Proliferation, and Apoptosis Analyses

Immunophenotypes of BM-PC and MRD analyses were performed in a minimum of 1 × 10^6^ or 2/3 × 10^6^ white cells, respectively, with FACSCanto-II and DIVA-Software (Becton Dickinson; BD; San Jose, CA, USA) following consensus criteria [4,46,47] (described in more detail in Figure 7). Briefly, total PCs were identified as CD38^+++^CD138^+/++^ events and aberrant PC as CD45^low/negative^ and/or CD19^low/negative^ and/or CD20^+^ and/or CD27^low/negative^ and/or CD56^+^ and/or monoclonal restriction for the heavy and/or light immunoglobulin chains. Mature B lymphocytes were defined as low FSC/SSC CD19^+^CD45^++^CD38^−/dim^ events.

The proliferation rate of aberrant PC was estimated as the percentage of CD38/CD138^+^ cells in the Synthesis + G2/M phases of the cell cycle by using Cycloscope-MM (Cytognos, Salamanca, Spain). Apoptosis rate of aberrant PC was estimated as the percentage of Anexin-V^+^ PCs minus the percentage of Anexin-V^+^ mature B lymphocytes by using anti-Anexin-V V450 (BD).

Given that both proliferation and apoptosis rates of BM-PC are functional biological properties that can be compromised by the time elapsed from extraction to laboratory processing, the transport, and/or storage conditions, we compared the Ratio-PA values and its prognostic capacity in in-house samples (analyzed in less than 2 h, *n* = 293), and in external samples which were processed in less than 4 h (*n* = 173) or 24 h (*n* = 142), depending on whether samples were shipped on the same day or the day after the BM extraction. BM samples were kept at room temperature and protected from extreme temperatures during transport. As described above, the apoptosis rate of mature B lymphocytes was extracted from the PC apoptosis rate as an internal control so as to minimize the impact of process times.

### 4.3. Fluorescent In Situ Hybridization (FISH)

Cytogenetic abnormalities were evaluated in an interphase nucleus from BM-PCs purified using a RosetteSep^®^ Human Multiple-Myeloma-Cell Enrichment Cocktail (Stemcell Technologies, Grenoble, France). The following FISH probes from Metasystems (Altlussheim, Germany) were used to evaluate: translocations of the immunoglobulin heavy chain gene region (IGH) with break-apart IGH probe (cut-off: 3%) and dual fusion probes to determine the most common IGH partners CCND1 (cut-off: 2%), FGFR3 (cut-off: 2%), MAF (cut-off: 2%), and MAFB (cut-off: 2%); the copy number of chromosomes 5, 9, and 15 with 5p15/9q22/15q22 hyperdiploidy probes (cut-off: 10%); amplification/deletion of 17p13 (TP53) and 17q22 (LPO/MPO) with locus-specific probes (cut-off: 10%); amplification/deletion of 1q21-22 (CKS1B) and 1p32.3 (CDKN2C) with locus-specific probes (cut-off: 10%), and monosomy-13/deletion 13q14.2 (DLEU1) and 13q34 (LAMP1) with locus-specific probes (cut-off: 10%). For each probe, 300 plasma cells were analyzed with Metafer (Metasystems).

### 4.4. Statistical Analysis

Statistical analyses were performed using SPSS version 15.0 (SPSS Inc, Chicago, IL, USA). Analysis of variance (ANOVA) and least significant difference (LSD) post hoc tests were used to analyze continuous variables. The receiver operating characteristic (ROC) was used to explore patient PFS and to determine the optimal cutoff values for proliferation and apoptosis rates, as well as for the proliferation-to-apoptosis ratio (Ratio-PA). PFS was estimated as months from the diagnosis date to disease progression or death. Survival curves were plotted according to the Kaplan-Meier method. The log-rank test was used to estimate significant differences. Multivariate analyses of prognostic factors for PFS and OS were performed using the Cox proportional hazards model (stepwise regression). The hazard ratio (HR) and 95% confidence interval were estimated. The Harrell C-statistic was obtained using STATA-14 (Somersd package, College Station, TX, USA). *p* < 0.05 was considered statistically significant.

## Figures and Tables

**Figure 1 ijms-22-03895-f001:**
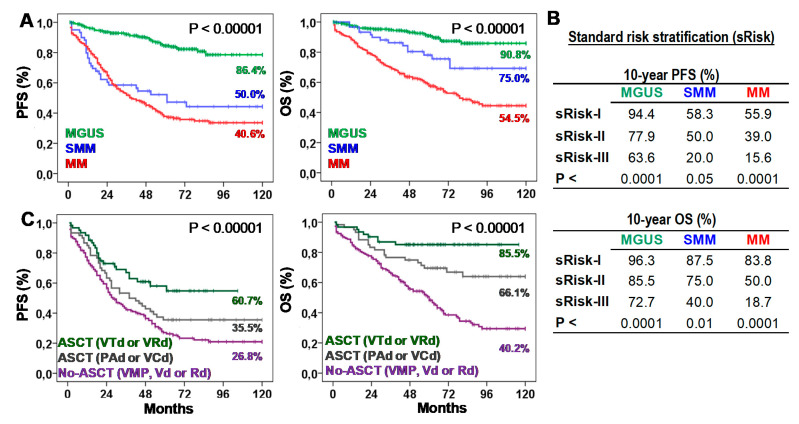
Survival of patients according to the type of plasma cell neoplasm (PCN), their standard risk or the type of first-line treatment. (**A**) Kaplan-Meier and Log-rank tests for Progression-Free (PFS) and Overall Survival (OS) according to the type of PCN: monoclonal gammopathy of undetermined significance (MGUS), smoldering multiple myeloma (SMM), or multiple myeloma (MM). Ten-year survivals are shown for each PCN type. (**B**) Ten-year PFS and OS rates (estimated with Kaplan-Meier and Log-rank tests) according to the standard risk (sRisk) for MGUS [30], SMM [7], and MM [8] patients: low (sRisk-I), intermediate (sRisk-II), and high (sRisk-III). (**C**) Kaplan-Meier and Log-rank tests for PFS and OS according to the type of first-line treatment: No-autologous stem cell transplantation (ASCT) with VMP, Vd or Rd; ASCT with PAD or VCD; or ASCT with VTd or VRd. Ten-year survivals are shown for each treatment.

**Figure 2 ijms-22-03895-f002:**
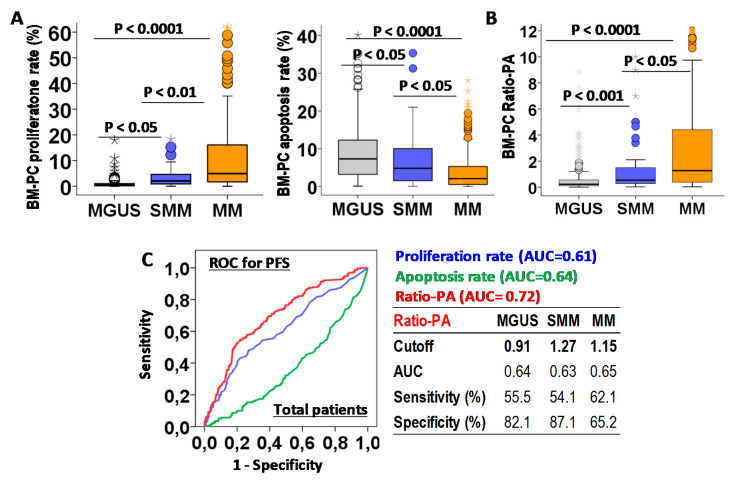
Biological characteristic of bone marrow plasma cells (BM-PCs) in patients with monoclonal gammopathy of undetermined significance (MGUS), smoldering multiple myeloma (SMM), or multiple myeloma (MM) and their prognostic capacity for progression-free survival (PFS)**.** (**A**) Quantification of BM-PCs proliferation and apoptosis rates in MGUS, SMM, and MM patients. (**B**) Proliferation-to-apoptosis ratio (Ratio-PA) of BM-PC in MGUS, SMM, and MM. (**C**) Receiver-operating characteristic (ROC) analysis of proliferation and apoptosis rates and Ratio-PA for PFS in total patients. Table shows cutoff values with maximum prognostic capacity for PFS, area under the curve (AUC), sensibility, and specificity of Ratio-PA in MGUS, SMM, and MM patients.

**Figure 3 ijms-22-03895-f003:**
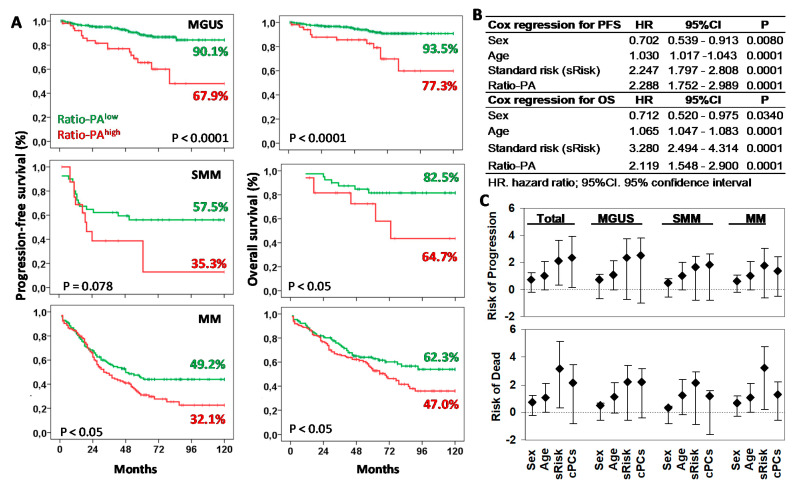
Prognostic capacity of Ratio-PA^high^ for progression-free survival (PFS) and overall survival (OS) of plasma cell neoplasm (PCN) stages. (**A**) Kaplan-Meier and Log-rank tests for PFS and OS according to the high or low Ratio-PA in patients with monoclonal gammopathy of undetermined significance (MGUS), smoldering multiple myeloma (SMM), or multiple myeloma (MM). Ten-year PFS and OS are indicated in the plots. (**B**) Cox regression analysis for PFS and OS for sex, age, standard risk stratification (sRisk), and Ratio-PA of bone marrow plasma cells in total patients. (**C**) Hazard ratio (HR) and 95% confidence interval for progression or death in total, MGUS, SMM, and MM patients observed in the Cox regression analysis.

**Figure 4 ijms-22-03895-f004:**
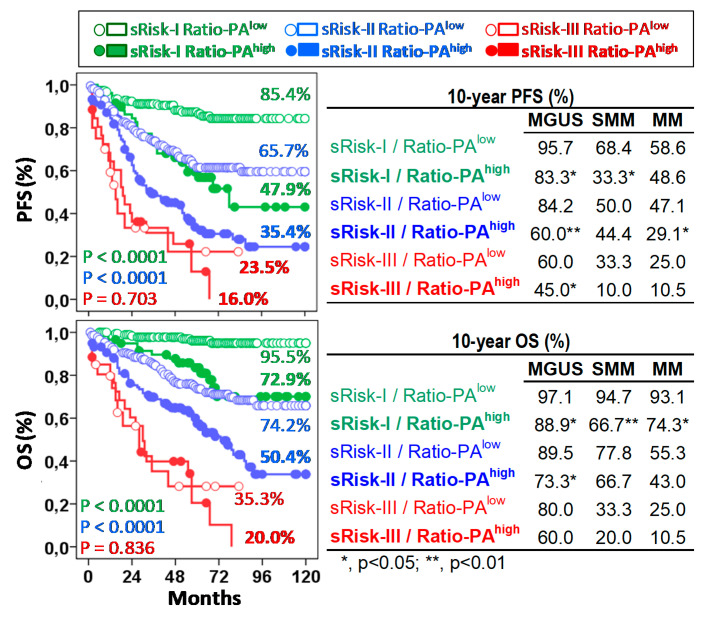
The Ratio-PA of bone marrow plasma cells complements standard risk stratification (sRisk). Left, Kaplan-Meier, and Log-rank tests for progression-free survival (PFS) and overall-survival (OS) of total patients according to Ratio-PA (high or low) in the standard risk groups (Risk-I, -II, and -III). Ten-year PFS and OS rates are indicated for each risk group in the plots. Tables on the right show same information for patients with monoclonal gammopathy of undetermined significance (MGUS), smoldering multiple myeloma (SMM), or multiple myeloma (MM). P estimated in the Log-rank test.

**Figure 5 ijms-22-03895-f005:**
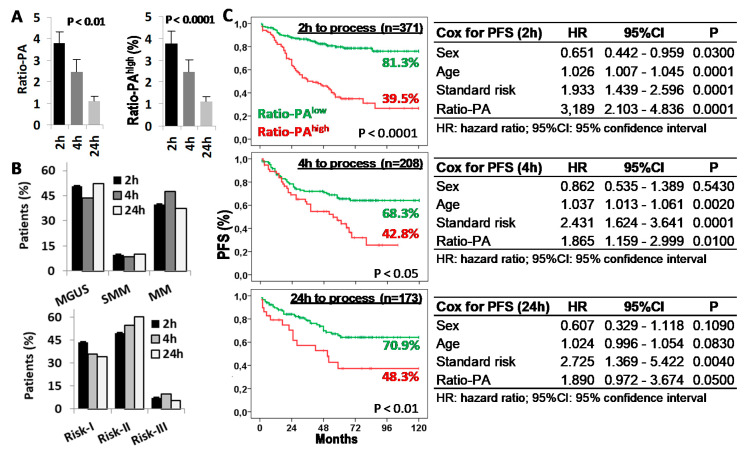
Time from bone marrow (BM) extraction to laboratory processing reduces the prognostic capacity of the Ratio-PA. (**A**) Ratio-PA of BM plasma cells and proportion of patients with Ratio-PA^high^ for samples processed 2 h, 4 h, or 24 h after BM extraction. (**B**) Distribution of patients by the plasma cell neoplasm (PCN) stage and the standard risk (Risk) group according to the processing time. (**C**) Kaplan-Meier and Log-rank tests for progression-free survival (PFS) according to Ratio-PA (high/low) in total PCN patients and according to the time until processing; and Cox regression analysis of PFS for sex, age, standard risk stratification, and Ratio-PA for samples processed 2 h, 4 h, or 24 h after BM extraction.

**Figure 6 ijms-22-03895-f006:**
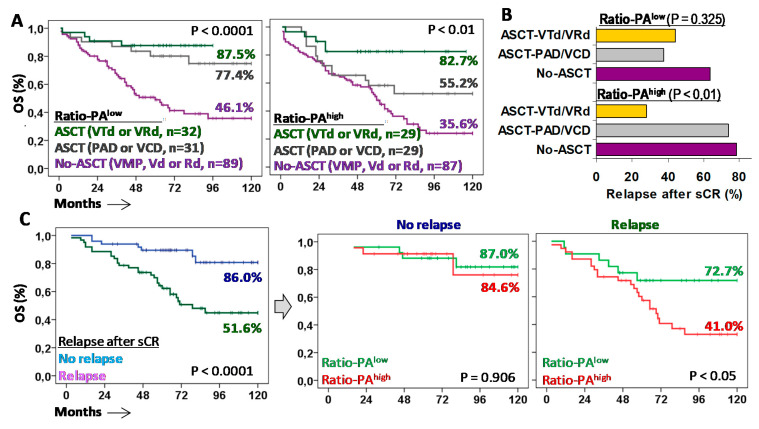
Autologous stem cell transplantation (ASCT) with VTd or VRd is the only effective first-line treatment for MM patients with Ratio-PA^high^. (**A**) Kaplan-Meier and Log-rank tests for overall survival (OS) for patients with low or high Ratio-PA according to the type of first-line treatment: No-ASCT with VMP, Vd or Rd; ASCT with PAD or VCD; or ASCT with VTd or VRd. (**B**) Percentage of patients who relapsed after stringent complete response (sCR) with negative minimal residual disease (MDR-) for low or high Ratio-PA and the three types of treatments. (**C**) Kaplan-Meier and Log-rank tests for OS according to relapse after sCR, and according to the Ratio-PA.

**Figure 7 ijms-22-03895-f007:**
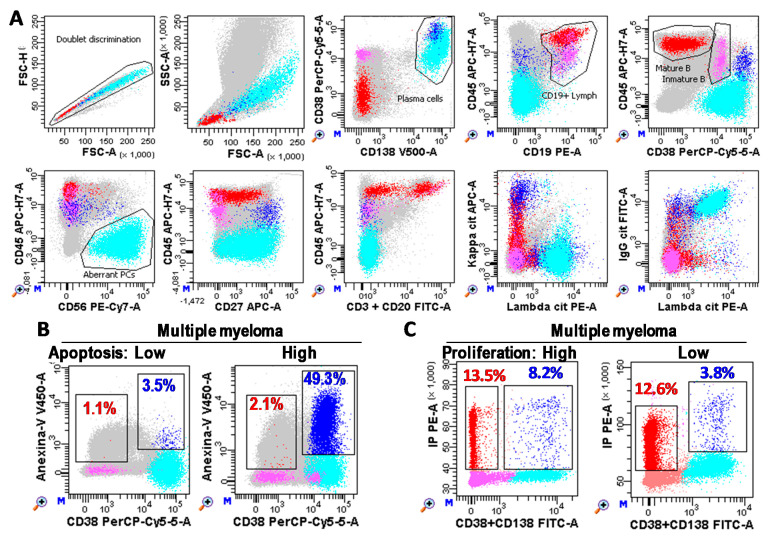
Plasma cell (PC) immunophenotyping. (**A**) Flow cytometry analysis of bone marrow samples performed with FACSCanto-II and DIVA Software (Becton Dickinson, San Jose, CA, USA). Photomultiplier (PMT) voltages were adjusted daily using CS&T beads (BD). Fluorescence compensations were finely adjusted using negative events as reference for each fluorochrome. A total of three million white cells were stained for each tube, tube-1: CD3&CD20 FITC, CD19 PE, CD38 PerCP-Cy5.5, CD56 PE-Cy7, CD27 APC, CD45 APC-Cy7, Annexin-V V450, and CD138 BV510 (BD); and tube-2: cyIgG, cyIgA, cyIgD, or cyIgM FITC, cyLambda PE, CD38 PerCP-Cy5.5, CD56 PE-Cy7, cyKappa APC, CD45 APC-Cy7, CD19 BV421, and CD138 BV510 (BD). One million cells were recorded for each tube. After doublet discrimination (in a FSC-H/SSC-A dotplot), total PC (Blue) were identified as CD38^+++^CD138^+/++^ events. Aberrant PC (cyan) were identified as CD45^low/negative^ and/or CD19^low/negative^ and/or CD20^+^ and/or CD27^low/negative^ and/or CD56^+^ and/or monoclonal restriction for the heavy and/or light immunoglobulin chains (specific gating strategy was follow for each patient based on their phenotype). Mature B cells (red) were identified as lymphocytes (FSC/SSClow) CD19^+^CD45^++^CD38^−/+^low. Immature B lymphocytes (pink) were defined as CD19^+^CD45^low^CD38^++^. Grey cells are non-B non-PC cells. (**B**,**C**) Bone marrow PCs (CD38^+^CD138^+^) from MM patients with low and high apoptosis (Anexin-V^+^ blue and red events for PCs and B lymphocytes, respectively) and proliferation rates (S + G2/M phases of the cell cycle), respectively.

**Table 1 ijms-22-03895-t001:** Baseline characteristics of patients at the time of diagnosis.

	MGUS (*n* = 316)	SMM (*n* = 57)	MM (*n* = 266)
**Demographic, Biochemical and Immunological Characteristics**			
Age, years, Mean ± SEM	67.4 ± 0.7	67.5 ± 1.6	67.6 ± 0.7
Female, *n* (%)	143 (45.3%)	34 (59.6%)	130 (48.9%)
Hemoglobin, g/dL, Mean ± SEM	14.3 ± 5.7	12.9 ± 2.51	10.8 ± 1.7 ***
Serum calcium, g/dL, Mean ±S EM	9.45 ± 0.04	9.46 ± 0.11	9.70 ± 0.09 **
Serum creatinine, mg/dL, Mean ± SEM	1.20 ± 0.07	1.04 ± 0.07	1.73 ± 0.13 *
Serum albumin < 3.5 g/dL, *n* (%)	29 (9.2%)	6 (10.5%)	91 (34.2%) ***
Serum 2-microglobulin ≥ 3.5 mg/dL, *n* (%)	93 (29.4%)	18 (31.5%)	159 (59.8%) ***
LDH ≥ upper limit of normal, *n* (%)	52 (16.4%)	9 (15.7%)	59 (22.2%)
Serum M-protein, g/dL, Mean ± SEM	1.04 ± 0.08	1.68 ± 0.13 *	2.81 ± 0.20 ***
Bence Jones protein, *n* (%)	106 (33.5%)	30 (52.6%) *	186 (69.9%) ***
Free light chain ratio > 20, *n* (%)	109 (34.5%)	32 (56.1%) **	195 (73.3%) ***
IgG gammopathy, *n* (%)	227 (71.8%)	33 (57.9%) *	143 (53.7%) ***
Immunoparesis, *n* (%)	111 (35.1%)	35 (61.4%) ***	242 (90.9%) ***
**Bone Marrow Plasma Cells (BM-PC) Counts**			
Total BM-PC histology, % (Mean ± SEM)	4.71 ± 0.16	16.85 ± 1.5 ***	33.52 ± 1.6 ***
Total BM-PC flow cytometry, % (Mean ± SEM)	1.08 ± 0.09	3.54 ± 0.54	13.01 ± 1.5 ***
**Fluorescent In Situ Hybridization (FISH) on Purified BM-PCs**			
del(17p), *n* (%)	3 (0.9%)	2 (3.5%)	23 (8.6%) ***
t(4;14) or t(14;16), *n* (%)	2 (0.7%)	2 (3.5%)	14 (5.2%) **
Gain of 1q21	35 (11.0%)	17 (30.0%) **	115 (43.2%) ***
Other alterations, *n* (%) ^1^	32 (10.1%)	7 (12.3%)	51 (20.3%) **
No abnormalities, *n* (%)	253 (80.1%)	30 (52.6%) **	77 (28.9%) ***
Not available, *n* (%)	55 (17.4%)	1 (1.75%) *	35 (13.15%)
**Clinical Characteristics**			
Osteolytic lesions, *n* (%)	8 (2.5%)	4 (7.0%)	147 (55.3%) ***
Renal insufficiency, *n* (%)	85 (26.9%)	14 (24.50%)	97 (36.4%) *
Additional cardio-respiratory diseases, *n* (%)	81 (25.6%)	15 (26.3%)	80 (30.1%)
Additional endocrine diseases, *n* (%)	72 (22.8%)	16 (28.1%)	63 (23.7%)
Additional rheumatologic diseases, *n* (%)	34 (10.7%)	4 (7.01%)	12 (4.5%)
Additional oncological malignances, *n* (%)	23 (7.2%)	7 (12.3%)	25 (10.1%)
Additional hematological diseases, *n* (%)	14 (5.1%)	5 (8.9%)	27 (10.3%)
Risk stratification Low/Intermediate/High, *n* ^2^	152/142/11	26/28/6	65/165/32
**Treatments ^3^**			
No ASCT with VMP, Vd or Rd, *n* (%)	17 (5.3%) ^4^	10 (17.5%) ^4^	149 (56.1%)
ASCT with PAD or VCD, *n* (%)	0 (0.0%)	3 (5.3%) ^4^	57 (21.4%)
ASCT with VTd or VRd, *n* (%)	2 (0.6%) ^4^	11 (19.3%) ^4^	48 (18.0%)
Palliative	-	-	12 (4.5%)

^1^ del(13q), other IGH translocations, hyper- or hypo-diploidy on Chromosome 5, 9, 13, 14, 15 or 17. ^2^ Risk stratification following standardized criteria for MGUS and SMM (score—0 = low, 1 = intermediate, and 2 = high) [7,30] and MM (RISS—I = low, II = intermediate and III = high) [8]. ^3^ ASCT: autologous stem cell transplantation; A: doxorubicin; C: cyclophosphamide; d: low-dose dexamethasone; M: melphalan, P: prednisone; V: bortezomib; R: lenalidomide; T: thalidomide. ^4^ MGUS and SMM patients who progressed to symptomatic PCNs and required treatment during the follow-up. * *p* < 0.05; ** *p* < 0.01 and *** *p* < 0.001 MGUS vs. SMM or MGUS vs. MM (ANOVA and LSD tests or Chi square test).

## Data Availability

Not applicable.

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
