# Peer review of "Proliferation to Apoptosis Tumor Cell Ratio as a Biomarker to Improve Clinical Management of Pre-Malignant and Symptomatic Plasma Cell Neoplasms"

_ijms, 2021, doi:10.3390/ijms22083895_

Round 1
Reviewer 1 Report
I was very interested to read your research. It seems quite logical that a high PA-ratio would be a more aggressive form of MM, and that the newer immunomodulants are more effective in this case, but I believe this research proves this concept. I think it's certainly of value in the process of decision-making of when to start treatment for SMM.
A question one might ask is when a patient presents with end-stage kidney disease, in which case there is no option to use neither thalidomide nor lenalidomide. Do you have any idea how many patients presented this way? In other words, in how many patients was the use of VTd or VRd contra-indicated and did the doctors have to resign to using VCD for the induction treatment?
Do you have any idea why the immunomodulants are more effective than the VCD or PAD treatment? In VCD and PAD, there are classical chemotherapeutics, which target mostly dividing cells. Why is it then, that immunomodulants are still more effective, even in cells which are highly proliferating?
Finally, it would be interesting to see the effect of adding a CD38-mAb to the treated groups, after a surveillance period of about five years (which is when the OS difference between the groups already begins to show in most cases)
Author Response
I was very interested to read your research. It seems quite logical that a high PA-ratio would be a more aggressive form of MM, and that the newer immunomodulants are more effective in this case, but I believe this research proves this concept. I think it's certainly of value in the process of decision-making of when to start treatment for SMM.
Thank you very much for your comments.
A question one might ask is when a patient presents with end-stage kidney disease, in which case there is no option to use neither thalidomide nor lenalidomide. Do you have any idea how many patients presented this way? In other words, in how many patients was the use of VTd or VRd contra-indicated and did the doctors have to resign to using VCD for the induction treatment?
In table-2, second line of “clinical characteristics”, you can read:
MGUS SMM MM
Renal insufficiency, n (%): 85 (26.9%) 14 (24.50%) 97 (36.4%)
In our records we have found that 43% of patients included in the non-eligible ASCT group treated with VMP, Vd or Rd that have renal insufficiency, 4 MGUS, 4 SSM and 64 MM.
None of the 3 SMM patients that receive ASCT with PAD and VCE have renal insufficiency. However, 23% (n=61) of MM patients that receive this treatment had renal insufficiency.
Do you have any idea why the immunomodulants are more effective than the VCD or PAD treatment? In VCD and PAD, there are classical chemotherapeutics, which target mostly dividing cells. Why is it then, that immunomodulants are still more effective, even in cells which are highly proliferating?
We do not have a clear answer to this question, but as you can see in figure-6B both types of treatments associated with ASCT reach similar stringent CR rates (≈66%); however, biological relapse rate was higher in PAD/VCD than in VTd/VRd (57.1% vs. 34.9%, p<0.05), so it could be a question of drug resistance of residual tumor cells. Given that the Ratio-RA is a two-component equation, it is possible that although the PAD/VCD has a strong influence on the proliferation component, the VTd/VRd is influencing both terms in the equations, inducing higher apoptosis and favoring less resistance and less relapse. We have added this idea in the 2nd paragraph of the discussion: “In fact immunomodulatory therapies base their anti-myeloma action on the induction of PC apoptosis, which could help to explain why ASCT eligible patients treated with IMiDs showed the best outcome in our series.22,23”
It has also been proposed that immunomodulants may improve the immune surveillance of T cell and NK cells. In fact we have described that genotype background of molecules regulating the function of both T and NK cells strongly influence the survival of myeloma (PMID: 27141379, this work was done with the first 150 cases of this manuscript), childhood acute leukemia (submitted) and solid cancer (PMID: 33076479, 30925758, 31411976, 31239317, and 30242020).
Finally, it would be interesting to see the effect of adding a CD38-mAb to the treated groups, after a surveillance period of about five years (which is when the OS difference between the groups already begins to show in most cases).
I agree that it would be really interesting to do such intervention, but this should be addressed in a formal clinical trial, I guess so?
Reviewer 2 Report
In this study, Vasco-Mogorrón et al. have applied the proliferation-to-apoptosis ratio (Ratio-PA) for risk stratification of plasma cell neoplasia showing a potential benefit of this marker in identifying patients with indolent disease (MGUS) at higher risk of progression to multiple myeloma (MM) or MM patients with poorer prognosis. The topic is very interesting and well introduced; however, there are some issues that should be addressed.
Major comments
The title is misleading and just focuses on one small aspect of this study which is also not well defined in the manuscript. Please consider to rephrase it as the manuscript describes a possible prognostic role of ratio-PA in plasma cell disorders.
The major concern of this study is the very short time required for processing (<2 hours from collection). The Authors should consider moving the paragraph 2.4 to the methods section and discuss how samples have been kept when acquired in 4 or 24 hours (room temperature? 4°C?). However, the Authors should thoroughly discuss why they did not validate the methodology on same in-house samples acquired at different timepoints (2-4-24 hours). Indeed, in that way, the Authors would have provided a time 0 and, using data from 4 and 24 hours, the % of loss and if this loss was significant or not. Or they might have used an “internal control” in each acquisition for standardization. In the way the Authors have validated their method, there are some biases, such as the physiological increased in apoptosis from collection and the fact that association between ratios and outcomes are patient-dependent. The Authors should perform ROC analysis (as in paragraph 2.2) also in this setting of different time-to-process, and try to apply the ratio-PA cut-off obtained from 4h or 24h processing to all samples and see if the ratio is still a good prognosticator. Otherwise, it is difficult to apply this methodology to routinely clinical practice as 2h is really a very short time.
What are sensibility and specificity of Ratio-PA when applying the MM cut-off (1.15) to all groups or the MGUS cut-off (0.91)?
On lines 238-239, I would have expected at least some differences especially in those who did not relapse. Therefore, only ratio-PA as prognostic marker might be not sufficient for risk stratification. Have the Authors considered to try the combination of R-ISS score and ratio-PA for stratifying patients according to treatment and relapse? Please discuss this point.
The Authors are focusing too much on therapeutic strategies and prognostic impact of ratio-PA in plasma cell disorders; however, ratio-PA alone seems not sufficient for risk stratification, and therapeutic decision is driven by other clinical features and performance status that allow a patient to undergo ASCT. Therefore, a patient who did not receive an ASCT might be a frailer or older subject and outcomes might be influenced by these clinical conditions. Indeed, patients with either ratio-PA high or low who received VTd/VRd+ASCT showed similar OS, as well as patients who did not receive the ASCT.
Samples were obtained at diagnosis and at 3 and 6 months of treatment. However, no mention about variations of ratio-PA at 3 and 6 months of therapy is described in the manuscript, and its correlation with clinical outcomes. Moreover, it would have been interesting showing variations of ratio-PA at disease progression.
The time-dependent variability of ratio-PA measurement is a great issue that makes difficult to standardize this marker in clinical practice. This is a true limitation of the study and should not be minimized in discussion (lines 277-285), while the Authors should address how to improve it also referring to published literature for other markers.
Minor comments
Several abbreviations are reported in the last paragraph of the manuscript.
The Authors should remove catalogue numbers of reagents (e.g., FISH probes).
P value annotation is not reported correctly. Please avoid annotations like P=1.1x10-12 and prefer P < 0.0001. In addition, in Figure 3B and 5C, some P values are reported as 0.000. Please change as P < 0.05 or P <0.001.
ANOVA should be performed between clinical characteristics reported in Table 1 to exclude differences in demographics among groups. P values should be reported.
Please maintain abbreviations in Tables and describe them in the caption (e.g., Bone marrow plasma cells (BM-PC) in Table 1) as for others. Keep Tables in the same page.
Please prefer darker colors rather than very light ones in graphs (e.g., light yellow in Figure 1C or light purple in Figure 6C).
Consider converting Figure 1B into Kaplain-Meier graphs.
Consider removing Figure 2A and 2D and adjusting graph proportions.
On Figure 4, consider using Kaplan-Meier graphs based on diseases instead of showing those bar graphs which are not a conventional way to indicate OS and PFS data.
Please be consistent with annotations. If you use comma for decimals (which is the European way), you should keep it throughout text and figures.
Lines 261-262, please prefer the following annotation style “mean+SD, 55.5+30.9 months”.

Author Response
In this study, Vasco-Mogorrón et al. have applied the proliferation-to-apoptosis ratio (Ratio-PA) for risk stratification of plasma cell neoplasia showing a potential benefit of this marker in identifying patients with indolent disease (MGUS) at higher risk of progression to multiple myeloma (MM) or MM patients with poorer prognosis. The topic is very interesting and well introduced; however, there are some issues that should be addressed.
Major comments
The title is misleading and just focuses on one small aspect of this study which is also not well defined in the manuscript. Please consider to rephrase it as the manuscript describes a possible prognostic role of ratio-PA in plasma cell disorders.
Thank you for your advice. This was our second option for the title “Proliferation to apoptosis tumor cell ratio as a biomarker to improve clinical management of pre-malignant and symptomatic plasma cell neoplasms”. You are right it is more appropriate to describe all results of the manuscript.
(1) The major concern of this study is the very short time required for processing (<2 hours from collection). (2) The Authors should consider moving the paragraph 2.4 to the methods section and discuss how samples have been kept when acquired in 4 or 24 hours (room temperature? 4°C?). (3) However, the Authors should thoroughly discuss why they did not validate the methodology on same in-house samples acquired at different timepoints (2-4-24 hours). Indeed, in that way, the Authors would have provided a time 0 and, using data from 4 and 24 hours, the % of loss and if this loss was significant or not. (4) Or they might have used an “internal control” in each acquisition for standardization. In the way the Authors have validated their method, there are some biases, such as the physiological increased in apoptosis from collection and the fact that association between ratios and outcomes are patient-dependent.
1- We fully agree that sample processing time should be one of our main concerns. Not only to analyze parameters such as proliferation and apoptosis, but also to carry out minimal residual disease (death of residual PC in 24 hours) and even immunophenotyping studies (some molecules can be modulated). However, logistically it is not easy to be able to analyze all samples in 2 hours, especially from peripheral hospitals, and for this reason, we believe that more than a concern, results described in our manuscript should be a warning showing that some parameters must be interpreted with caution when they are analyzed in samples with more than 24h. This is the main reason why we decided to include this section in the manuscript, even knowing that it might be a concern for reviewers!
2- We have moved information from paragraph 2.4 to the method section, and describe the way to conserve samples till processing.
3- This was a retrospective study in real life medicine patients performed collecting the flow cytometry and clinical information from our records. We introduced proliferation analysis at diagnosis of new patients in 2009 to help to estimate the risk in MGUS and SMM patients (https://doi.org/10.1182/blood-2007-05-088443), but extended the study to all PCN stages since we did not always know the stage of the patient when receiving the sample. The study of apoptosis was initiated to evaluate if this parameter could have any predictive value with the generalized introduction of IMIDs at this time. That is why we have these tow parameters evaluated homogeneously in so many patients for a 10 year period. I agree that it could be really interesting to evaluate proliferation and apoptosis at different time-points of the same samples. But then, the value with the best predictive capacity should be estimated in new samples and new patients and it will take years to have a clear answer. However, by comparing patients and samples from different hospital we can some-how extrapolate that conclusion, can`t we? I our opinion the conclusion of this section should be that samples need to be analyzed as soon as possible. In fact, hospitals in our region that were sending samples with 24h have changed their procedures to send the samples in the same day of the extraction to be evaluated in less than 4 hours.
4- In fact, we did use an internal control which was the apoptosis rate of mature B lymphocytes that was subtracted from the PC apoptosis rate. We have indicated this fact more clearly in the method section.
The Authors should perform ROC analysis (as in paragraph 2.2) also in this setting of different time-to-process, and try to apply the ratio-PA cut-off obtained from 4h or 24h processing to all samples and see if the ratio is still a good prognosticator. Otherwise, it is difficult to apply this methodology to routinely clinical practice as 2h is really a very short time.
When we found that the time to processing was influencing Ratio-PA in this way, we investigated if we could find a cut-off that would allow us to better classify each of these patients. However, due to the small number of MM cases for 4h (n = 80) and 24h (n = 50) the ROC curve calculations were not so precise, and we decided to maintain a compromised value useful for all samples (and estimated with all samples). In fact, the independent predictive value of Ratio-PA estimated that way remained valid even in samples that were analyzed 24 hours after extraction.
What are sensibility and specificity of Ratio-PA when applying the MM cut-off (1.15) to all groups or the MGUS cut-off (0.91)?
MGUS (cutoff=0.91, sensitivity=55.5% and specificity=82.1%) - cut-off=1.15, S=56% and E=54%
SMM (cutoff=1.27, sensitivity=51.1% and specificity=87.1%) - cut-off=1.15, S=29% and E=87%
MM (cutoff=1.15, sensitivity=62.5% and specificity=65.2%) - cut-off=0.91 S=26% and E=91%
On lines 238-239, I would have expected at least some differences especially in those who did not relapse. (1) Therefore, only ratio-PA as prognostic marker might be not sufficient for risk stratification. (2) Have the Authors considered to try the combination of R-ISS score and ratio-PA for stratifying patients according to treatment and relapse? Please discuss this point.
We have included a Supplementary figure-1 describing the impact of the Ratio-PA in patients with different RISS stages according to their eligibility for ASCT and the relapse status after stringent complete response. You were right, from this analysis we can now observe that patients that did not relapse after 3/6 month, with RISS-II or RISS-III and high Ratio-PA had worst outcome, but no differences were observed in those with RISS-I. In relapsed patients differences were clearly observed in patients with RISS-I and RISS-II (to few patients in RISS-III so see differences). We describe and discuss these new results.
The Authors are focusing too much on therapeutic strategies and prognostic impact of ratio-PA in plasma cell disorders; however, ratio-PA alone seems not sufficient for risk stratification, and therapeutic decision is driven by other clinical features and performance status that allow a patient to undergo ASCT. Therefore, a patient who did not receive an ASCT might be a frailer or older subject and outcomes might be influenced by these clinical conditions. Indeed, patients with either ratio-PA high or low who received VTd/VRd+ASCT showed similar OS, as well as patients who did not receive the ASCT.
You are right in your appreciation about not showing differences in the survival of patients with Ratio-PA high/low and ineligible for ASCT or treated with VTd/VRd+ASCT. That is way we conclude in the last paragraph of the discussion and in the abstract that “every effort should be made to provide first-line therapies including VTd or VRd associated with ASCT to patients with Ratio-PAhigh”. It is the only situation where Ratio-PA can play a decisive role in the decision making of patients requiring therapy. I know that clinical/biological conditions will not allow the introduction of IMIDs in all patients, but the clinician should know that if Ratio-PA is high outcome in these patients will be worst.
Besides, it should also be considered that in MGUS and SMM Ratio-PAhigh identified patients at higher risk of progression and dead, so these patients could have also benefited from advanced treatments, as discussed in the last two paragraphs of the discussion.
Samples were obtained at diagnosis and at 3 and 6 months of treatment. However, no mention about variations of ratio-PA at 3 and 6 months of therapy is described in the manuscript, and its correlation with clinical outcomes. Moreover, it would have been interesting showing variations of ratio-PA at disease progression.
Unfortunately, studding the impact of Ratio-PA variations during patient`s clinical course was not our objective, so proliferation and apoptosis data after treatment are not available. Besides, we only receive BM sample to perform MRD in patients that for clinical protocol or faculty decision required that study during the treatment. Anyway we have included in the manuscript a reference describing the utility of monitoring PC proliferation during the clinical course of the disease (https://doi.org/10.1182/blood-2007-05-088443). Nonetheless, this type of studies have important bias, since it can only be performed in a limited number of patients that have enough number of PC after intensive treatment and second it cannot be performed in patients that reach and maintain sCR.
The time-dependent variability of ratio-PA measurement is a great issue that makes difficult to standardize this marker in clinical practice. This is a true limitation of the study and should not be minimized in discussion (lines 277-285), while the Authors should address how to improve it also referring to published literature for other markers.
It is not our intention at all, to minimize this fact. On the contrary, we believe we are the first to describe this problem that may have gone unnoticed in other studies. For this reason we say in the discussion “Although, in light of our results, these studies may have underestimated the prognostic capacity of these parameters”. Besides, we emphasize this fact both in the results section and the discussion and in the final conclusion: “proliferation to apoptosis ratio (Ratio-PA) of PC, estimated at diagnosis by flow cytometry in the shortest possible time from BM extraction (preferably <2h), offers a prognostic biomarker…….". Of course that this might be a problem for standardization, that is way if any clinical trial in the future address the study of these parameters should take in consideration the information described in this manuscript to make every effort to evaluate these parameters in local laboratories in the shortest possible time.
Minor comments
Several abbreviations are reported in the last paragraph of the manuscript.
We are not sure if you mean the last paragraph of the method section: Statistical analysis. If so, we have defined ANOVA and change DMS for “least significant difference (LSD)”. (DMS was Spanish abbreviation in the SPSS program, sorry)
The Authors should remove catalogue numbers of reagents (e.g., FISH probes).
Ok, done.
P value annotation is not reported correctly. Please avoid annotations like P=1.1x10-12 and prefer P < 0.0001. In addition, in Figure 3B and 5C, some P values are reported as 0.000. Please change as P < 0.05 or P <0.001.
Ok, done.
ANOVA should be performed between clinical characteristics reported in Table 1 to exclude differences in demographics among groups. P values should be reported.
Ok, done.
Please maintain abbreviations in Tables and describe them in the caption (e.g., Bone marrow plasma cells (BM-PC) in Table 1) as for others. Keep Tables in the same page.
Ok. But I am not sure to have found all of them.
I am keeping the table-1 in two pages to respect the format established by editorial reviewers.
Please prefer darker colors rather than very light ones in graphs (e.g., light yellow in Figure 1C or light purple in Figure 6C).
Ok, done.
Consider converting Figure 1B into Kaplain-Meier graphs.
That would mean 6 Kaplan-Meier plots in a single sub-figure. It would be a little too dense.
I have substitute bar plots for a table with the 10y survival rates. Hope you prefer this format.
Consider removing Figure 2A and 2D and adjusting graph proportions.
We have deleted figure 2A since that information was repeated in table-1. We have deleted figure 2D but kept the information in the text, so we describe the proportion of Ratio-PA-high within each PCN stage.
On Figure 4, consider using Kaplan-Meier graphs based on diseases instead of showing those bar graphs which are not a conventional way to indicate OS and PFS data.
That would mean 6 Kaplan-Meier plots in a single sub-figure. It would be a little too dense.
I have substitute bar plots for a table with the 10y survival rates. Hope you prefer this format.
Please be consistent with annotations. If you use comma for decimals (which is the European way), you should keep it throughout text and figures.
I tried, but not succeed. Hope I have found all these errors now. Thanks
Lines 261-262, please prefer the following annotation style “mean+SD, 55.5+30.9 months”.
Ok, done.
Round 2
Reviewer 2 Report
The Authors have satisfactorly addressed all raised points. Thanks.